

# An all-sky camera images classification method using cloud cover features

Xiaotong Li, Baozhu Wang, Bo Qiu, and Chao Wu

School of Electronics and Information Engineering, Hebei University of Technology, Tianjin, 300401, China

*Correspondence to*: Bo Qiu (qiubo@hebut.edu.cn)

**Abstract.** The all-sky camera (ASC) images can reflect the local cloud cover information, and the cloud cover is one of the first factors considered for astronomical observatory site selection. Therefore, the realization of automatic classification of the ASC images plays an important role in astronomical observatory site selection. In this paper, three cloud cover features are proposed for the TMT (Thirty Meter Telescope) classification criteria, namely cloud weight, cloud area ratio and cloud

dispersion. After the features are quantified, four classifiers are used to recognize the classes of the images. Four classes of ASC images are identified: "Clear", "Inner", "Outer" and "Covered". The proposed method is evaluated on a large dataset, which contains 7328 ASC images taken by an all-sky camera located in Xinjiang (38.19° N, 74.53° E). In the end, the method achieves an accuracy of 97.28% and F1_score of 96.97% by a random forest (RF) classifier, which greatly improves the efficiency of automatic processing of the ASC images.

## 1 Introduction

Clouds are visible polymers composed of water vapor in the atmosphere liquefied by cold, and they are important for the hydrological cycle and energy balance of the Earth (Sodergren et al., 2017). The analysis of clouds can provide a lot of valuable information such as weather prediction (Westerhuis et al., 2020), climate prediction, astronomical observatory site selection (Cao et al., 2020) and so on. For astronomical observatory site selection, cloud cover is a factor that must be considered. When

the light emitted by a star reaches the telescope, it will be scattered and absorbed by clouds in the atmosphere. Therefore, the cloud cover determines the quality of the observation data and the available time for astronomy. Currently, cloud observations are mainly based on satellite remote sensing (Zhong et al., 2017; Young et al., 2018) and ground-based observations (Nouri et al., 2019). Satellite cloud images can capture large areas of cloud cover and directly observe the impact of clouds on the earth's radiation, which is suitable for atmospheric research. However, the resolution of the images is low, making it impossible to

study more cloud details. Ground-based cloud images have a smaller observation range and focus on monitoring the cloud thickness and distribution in local areas, which has the advantages of flexible observation sites and rich image information. This paper mainly focuses on the classification of ground-based cloud images.

The all-sky imaging device is an automatic observation instrument that can replace humans. The device has a digital camera that can capture images of half of the celestial sphere and can obtain details of clouds through retrieval algorithms. With the





30 emergence of more and more all-sky imaging devices such as Whole Sky Imager (WSI; Sneha et al., 2020), Total Sky Imager (TSI; Ryu et al., 2019), All Sky Imager (ASI; Nouri et al., 2018) and All Sky Camera (ASC; Fa et al., 2019), a large number of all-sky images have now been generated, which provides a basis for the development of automatic cloud images classification algorithms.

 Many researchers pay attention to the feature extraction technology of different cloud types. Calbo and Sabburg (2008)

35 classified eight types of sky images using statistical features (smoothness, standard deviation, uniformity) and features obtained after Fourier transform of the images. Heinle et al. (2010) proposed a classification algorithm based on spectral features in RGB color space and texture features extracted by grey-level co-occurrence matrices (GLCMs). This method has a high accuracy for the classification of seven different classes of clouds. Sun et al. (2009) proposed a method where the local binary pattern (LBP) operator and the contrast of local cloud image texture are used to classify sky conditions. Li et al. (2016)

40 expressed each image as a feature vector, which was generated by calculating the weighted frequency of the microstructures observed in the image. Dev et al. (2015) investigated an improved text-based classification method that combines color and texture features to improve the effect, the average accuracy on the Singapore Whole-Sky Imaging Categories (SWIMCAT) dataset is 95 %. Gan et al. (2017) worked on different size regions of the image and they used a method based on sparse coding which is effective in terms of localization and computation. Wan et al. (2020) demonstrated that cloud classification result was

45 not well if only texture or color features are used. As a result, a mixture of texture, color and spectral features were obtained from color images and then fed into a random forest (Svetnik et al., 2003).

 In recent years, convolutional neural network (CNN) has shown very superior effects in the field of image classification and have been applied to all-sky images classification. Shi et al. (2017) used the output of the shallow convolutional layer of CNN as cloud features and fed it into a support vector machine (SVM) (Cristianini and Shawe-Taylor, 2000). Ye et al. (2017)

50 extracted multiscale feature maps from pretrained CNN and then employed the Fisher vector to encode them, finally sent them to a classifier. Zhao et al. (2019) used the 3D-CNN model to extract the texture features of the images and then output the classification results with a fully connected layer. Zhao et al. (2020) proposed the improved Frame Difference Method extractor to detect and extract features from large images into small images, then these small images were sent to a multi-channel CNN classifier. Liu et al. (2021) proposed context graph attention network (CGAT) for the cloud classification, which the context

55 graph attention layer learned the context attention coefficients and acquired the aggregated features of nodes.

 The algorithms proposed above, both traditional and deep learning methods, are based on texture, color and spectral features of cloud for classification, and they are all for cloud shape classification of local images or all-sky images. But for the astronomical observatory site selection, cloud cover is also an important factor that must be considered besides cloud shape. At present, few researchers have studied the classification algorithm based on cloud cover. The existing cloud cover calculation

60 method only calculates the ratio of the cloud coverage area to the effective area of the ASC images, and does not provide a detailed description of the cloud cover (Esteves et al., 2021). Therefore, we propose three cloud cover features in this paper that introduces cloud thickness and distribution position into the cloud cover calculation, and classify the ASC images according to the TMT classification criteria. The rest of the paper is organized as follows. Section 2 introduces the dataset and



TMT classification criteria. Section 3 describes three cloud cover features and proposed classification method. Section 4 shows
65 experiment results and analysis of the results. Finally, it is our conclusion in Section 5.

## 2 Dataset and image classes

### 2.1 Dataset

Images used in this paper were taken by an all-sky camera located in Xinjiang, China (38.19° N, 74.53° E), and provided by
the Key Laboratory of Optical Astronomy at the National Astronomical Observatories of Chinese Academy of Sciences. The
70 ASC has two parts, a Sigma 4.5 mm fisheye lens and a Canon 700D camera providing a 180°×180° field of view. The frequency
of shooting in daytime is 20 minutes, increasing to 5 minutes a shot at night, and the exposure time will be adjusted between
15 and 30 seconds depending on the phase of the moon. The ASC images are stored in color JPEG format with a resolution of
480×720 pixels. In order to facilitate subsequent processing, the images are cropped to retain only the central valid area, and
the cropped image size is 370×370 pixels. Note that the captured image is originally rectangular but the mapped all sky is
circular, where the center is the zenith and the boundary is the horizon. Figure 1a shows an example of the original ASC image
and Fig. 1b displays the cropped ASC image.

We screened the ASC images taken from January 2019 to November 2020 to remove those that were overexposed or
encountered bad weather. The screened images are all classified by professionals using a similar manual cloud identification
method (see the next section) as described in the Thirty Meter Telescope (TMT) site testing campaign paper (Skidmore et al.,
2008). In the end, a dataset with 7328 ASC images was obtained. We did our best to ensure a balanced number of images of
each class in the dataset.

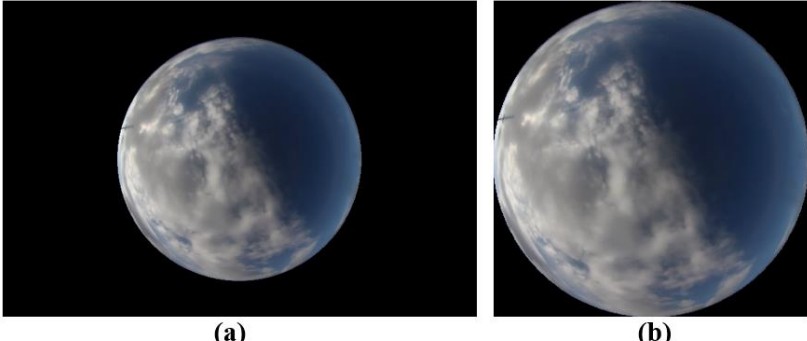

(a)  (b)

**Figure 1.** Example of ASC image. (**a**) Original RGB color ASC image, (**b**) Resized RGB color ASC image of (**a**).

### 2.2 Image classes

Traditionally, the ASC image classification takes cloud shape as the basic element, while considering the shape development
and internal microstructure of the clouds. However, the cloud cover in the image is the first important factor to be considered





for astronomical observatory site selection. Therefore, we classify the ASC images following the method of Skidmore et al. (TMT classification criteria; 2008). The ASC images are divided into inner and outer circles with zenith angles of 44.7° and 65° respectively, then the cloud cover is determined as "Clear", "Outer", "Inner" and "Covered" according to the thickness

and distribution of the cloud on the ASC images. The classification definition is shown in Table 1, and typical images of each class are shown in Fig. 2.

**Table1.** The ASC image classes and corresponding description.

| Class | Description |
| --- | --- |
| Clear | No thick cloud within the inner circle. Fig. 2a |
| Outer | No thick cloud within the inner circle, has cloud within the outer circle. Fig. 2b |
| Inner | No more than 50% cloud within both the inner and outer circles. Fig. 2c |
| Covered | Coverage of cloud within inner and outer circles is over 50%. Fig. 2d |

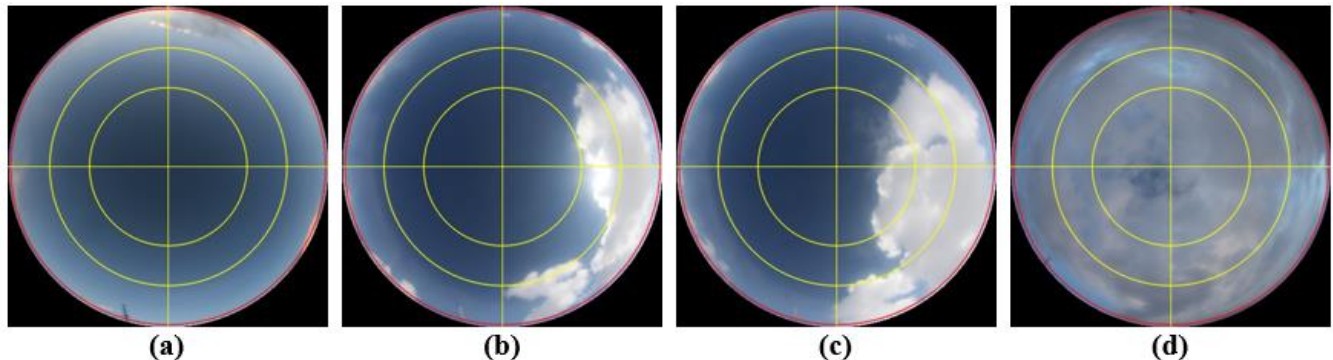

**Figure 2.** ASC image samples from dataset. (**a**) "Clear", (**b**) "Outer", (**c**) "Inner", (**d**) "Covered".

## 3 Classification based on cloud cover features

In this section, we first describe the cloud cover features proposed, and then introduce the overall process of the ASC images classification method based on the cloud cover features. The framework is illustrated in Fig. 3.

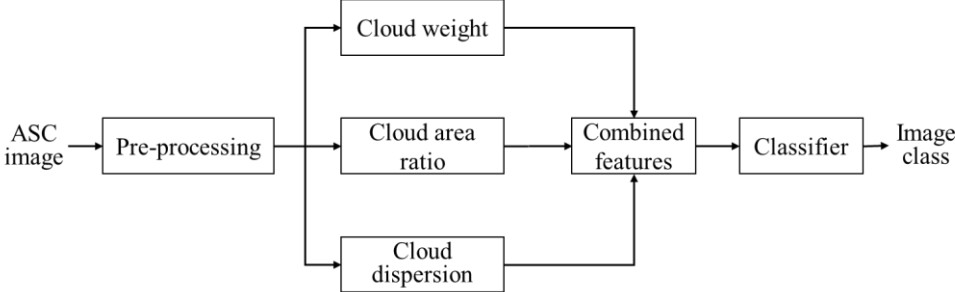

**Figure 3.** System framework.



### 3.1 Cloud cover features

The current TMT classification criteria is essentially only based on the area ratio and distribution of the cloud regions in the ASC images with insufficient consideration of cloud thickness and density of distribution, and no intuitive and reliable quantitative indicators, so the classification results are easily affected by human subjective factors. In this paper, three cloud
features are proposed for the above problems, namely cloud weight, cloud area ratio and cloud dispersion. Among them, cloud weight indicates the thickness of the cloud, cloud area ratio represents the ratio of the cloud area to the effective area of the image and cloud dispersion reflects the distribution of cloud regions around the center of the ASC images.

### 3.1.1 Cloud weight

The grayscale value $G$ of any pixel in the ASC image can be regarded as the weighted superposition value of the grayscale
value of multiple elements such as cloud, sky background and impurities in an ideal state, as shown in Eq. (1). The ideal state means that the grayscale value of the pixel is contributed by only one element and no other elements are involved, and the pixel can be regarded as the "pure pixel" of the element.

$$G = \alpha_1 G_1 + \alpha_2 G_2 + \cdots + \alpha_N G_N \tag{1}$$

where $G_1$, $G_2$, ..., $G_N$ denote the grayscale value of the "pure pixel" of different elements. $\alpha_1$, $\alpha_2$, ..., $\alpha_N$ are the weights of the
contribution of the "pure pixel" of each element to the actual grayscale value of the image pixel.

For an ASC image, the grayscale value of the cloud region ideally is superimposed by the grayscale value of the sky background and cloud. Therefore, the grayscale value of cloud region can be defined by the superimposed model as:

$$G = \alpha G_{sky} + (1 - \alpha) G_{cloud} \tag{2}$$

where $G$ is the true grayscale value of the cloud region; $G_{sky}$ is the grayscale value of the sky background in an ideal state.
Since the shooting time of the image is known, the grayscale value of the sky background can be obtained by using images of different dates but the same moment; $G_{cloud}$ is the grayscale value of cloud in the ideal state; $\alpha$ is the weight of the contribution of the sky background to the actual grayscale value of the pixel in the ideal state. During the shooting process of ASC, the sky background will be blocked by clouds. As the cloud becomes thicker and thicker, the contribution of the sky background to the grayscale value of cloud region will gradually decrease, and this phenomenon will be more obvious in the initial stage
when the cloud thickness increases. When the cloud thickness increases to a certain degree, the grayscale value of cloud region is almost completely contributed by the cloud, so the relational expression between $\alpha$ and $G_{cloud}$ is approximately a monotonically decreasing concave function. After extensive experimental verification, the relationship between the two is derived in this paper as:

$$\alpha = \frac{G_{sky}}{G_{cloud} + G_{sky}} \tag{3}$$

According to Eq. (2-3), $G_{cloud}$ is calculated as follows:





$$G_{cloud} = \frac{G + \sqrt{G^2 - 4(G_{sky} - G)G_{sky}}}{2} \qquad (4)$$

In order to verify the validity of the cloud grayscale value obtained by the superposition model, we process the grayscale image of the local cloud region according to Eq. (4), and the result is shown in Fig. 4. Figure 4a shows the original grayscale image, and Fig. 4b is the grayscale image in the ideal state obtained by using the superposition model. By comparison, it can be found

that the grayscale value of the processed image is significantly lower in the thin cloud part, which eliminates the influence of the sky background on the cloud grayscale value, thereby better reflecting the cloud contour. Therefore, the grayscale image processed by the superposition model reflects the grayscale value and the number of cloud pixels more realistically.

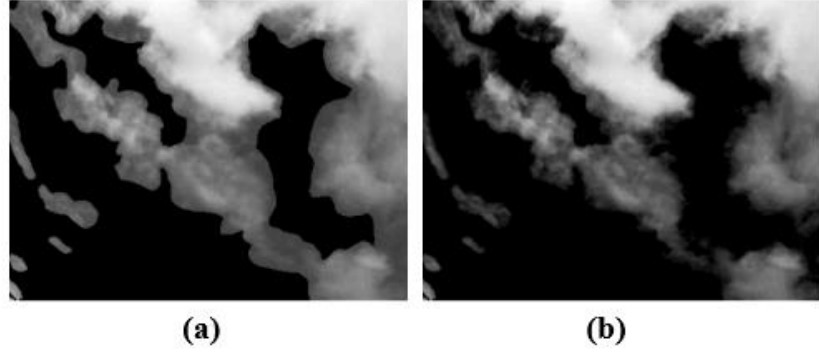

**Figure 4.** Result of superimposed model process. (**a**) Original grayscale image, (**b**) processed grayscale image.

Based on the previous derivation, we proposed "cloud weight" to indicate the thickness of the clouds. Since there are large differences in color and brightness between clouds and sky in ASC images, this section defines the cloud weight by exploring the relationship between pixel grayscale value, cloud reflectivity, light intensity and cloud thickness.

In grayscale images, the magnitude of the grayscale value is related to the reflectivity of the object and the intensity of the incident light. The grayscale value can represent the brightness of the pixel, which is determined by the reflectivity and the

intensity of the incident light. Therefore, the grayscale value of a cloud pixel $G_{cloud}$ is calculated as:

$$G_{cloud} = rl \qquad (5)$$

where $G_{cloud}$ indicates the grayscale value of the pixel in the ideal state, $r$ represents the reflectivity of the cloud to the light and $l$ is the intensity of the incident light. Generally speaking, the reflectivity of the cloud to the light $r$ is proportional to the cloud thickness $w$. The thicker the cloud, the higher the reflectivity, the expression between the two is approximately as:

$$r = \lambda w \qquad (6)$$

where $\lambda$ is the correlation coefficient. The incident light intensity $l$ is determined by the incident light and the incident light sources are the same for the cloud and sky background. Compared with cloud, the sky background is simpler and does not have an overly complex function change in mapping the light source onto the grayscale image, so the magnitude of the sky background grayscale value $G_{sky}$ is proportional to the intensity of the light source, then $G_{sky}$ is:

$$G_{sky} = \mu l \qquad (7)$$



where $\mu$ is the positive correlation coefficient. According to Eq. (5-7), $w$ is calculated as follows.

$$w = \frac{\mu G_{cloud}}{\lambda G_{sky}} \tag{8}$$

Since $\mu/\lambda$ is a constant and does not affect the classification result, we omit $\mu/\lambda$ in the subsequent calculations. Then the value

of $w$ represents the relative size of the cloud weight, which can approximately reflect the thickness of the cloud. In order to

facilitate subsequent calculations and comparisons, we normalized $w$ as follows.

$$w_n = \frac{w - \min(w)}{\max(w) - \min(w)} \tag{9}$$

where $w_n$ is the normalized value of cloud weigh, $min(w)$ and $max(w)$ represents the minimum and maximum values of

cloud weight in the cloud region of the image respectively.

This section defines the cloud weight based on the grayscale value to reflect the thickness of the cloud. The physical meaning

of cloud weight is the difference between the brightness of the cloud and sky background, which can approximate the thickness

of the cloud region. The feature of cloud weight makes the evaluation of cloud thickness completely based on grayscale images,

which promotes the automatic processing of ASC images.

### 3.1.2 Cloud area ratio

At present, the common cloud area ratio calculation methods include ISCCP (Evan et al., 2007), CLAVR-1 (Wang et al., 2013)

and CLAVR-X (Kim et al., 2016), etc. ISCCP divides pixels into non-cloud and cloud pixels and assigns weights of 0 and 1

respectively to calculate the ratio of the weighted sum of the two to the total number of pixels as the final result. CLAVR-1

divides the pixels into non-cloud, mixed cloud and cloud, then assigns weights of 0, 0.5, and 1 respectively to calculate the

ratio of cloud area. CLAVR-X classifies the image pixels into non-cloud, thin cloud, medium cloud and thick cloud according

to the cloud thickness, and the weights are adjusted according to the cloud thickness. All the above calculation models divide

the pixels into limited categories and assign weights respectively. Although cloud thickness and cloud coverage area are

considered as two cloud cover features, the division of cloud thickness is slightly rough. In the previous section, we completed

the preliminary exploration of cloud weight and obtained a more accurate and scientific numerical representation of cloud

thickness, so that the weight of cloud pixels can be optimized more accurately.

This work refines the weight for each pixel of the ASC images and redefines the calculation model of cloud area ratio. We use

the cloud weight derived in the previous section to represent the weight of cloud pixels and define the ratio of the weighted

sum of all pixels to the effective area of the image as the cloud area ratio (CAR), whose equation is:

$$CAR = \frac{\sum_{i=1}^{N_{cloud}} w_n^i}{N_{all}} \tag{10}$$

where $N_{cloud}$ is the total number of cloud pixels in the image, $w_n^i$ denotes the cloud weight of each cloud pixel and $N_{all}$

represents the total number of pixels in the effective area of the ASC image.



### 3.1.3 Cloud dispersion

Cloud dispersion (CD) is proposed mainly to indicate the influence degree of the distance between cloud and the center of the ASC image on astronomical observations results. The closer to the center, the greater the influence. Cloud dispersion is a quantitative representation of this effect. Generally speaking, cloud dispersion has three determinants: cloud area, cloud thickness and the distance between cloud and the center of the image. Therefore, we divide the image into $N$ regions, namely $x_i$ ($i = 1,2,...,N$), the area of $x_i$ is denoted as $s_i$, the average cloud thickness is denoted as $w_i$ and the absolute distance between the center of the region and the center of the ASC image is $d_i$. In order to facilitate the calculation, we convert the absolute distance to the relative distance $d_i^* = d_i/R$, where $R$ represents the radius of the effective area of the image. Then cloud dispersion of an ASC image can be represented using Eq. (12):

$$CD = \sum_{i=1}^{N} s_i w_i f(d_i^*) \qquad (11)$$

where $f(d_i^*)$ is an influence degree function with $d_i^*$ as the independent variable. The smaller the $d_i^*$, the greater the influence on the observation results, and the greater the value of $f(d_i^*)$. From the above, it can be seen that $f(d_i^*)$ is a function that decreases monotonously with $d_i^*$, and the influence on the observation results is greatest when the cloud is located at the image center. Since the classification of the ASC images in this paper is based on the TMT classification criteria, it is necessary to consider the influence of the inner and outer circle when setting the parameters. After many experiments and comparison studies with the results of manual observation, we set the value of influence degree at the center of the ASC image to 1, the inner circle to 0.7 and the outer circle to 0. After linear fitting by MATLAB curve fitting toolbox, $f(d_i^*)$ is shown in the Fig. 5, the expression is Eq. (13), where the sine term is to correct the fitted quadratic function to satisfy the decreasing condition.

$$f(d_i^*) = -24.33 \sin(d_i^* - \pi) + 1.228(d_i^* - 10)^2 - 121.8 \qquad (12)$$

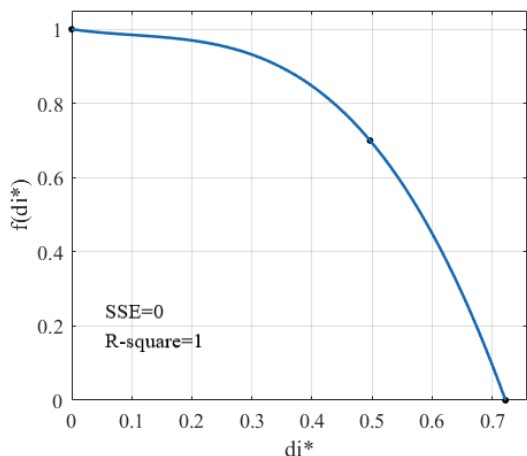

**Figure 5.** Curve fitting between $d_i^*$ and $f(d_i^*)$.



In actual calculations, this paper takes each pixel as the calculation unit, which can make the measurement of cloud dispersion accurate to the pixel level. Then the region $x_i$ represents the individual image pixel whose area $s_i$ is 1 and the cloud thickness $w_i$ can be expressed by the cloud weight in section 3.1.1. According to Eq. (12-13), the cloud dispersion calculation model based on pixel can be obtained as:

$$CD = \sum_{i=1}^{N_{cloud}} w_n^i [-24.33 \sin(d_i^* - \pi) + 1.228(d_i^* - 10)^2 - 121.8] \tag{13}$$

## 3.2 Classification of ASC images

### 3.2.1 Pre-processing

Since the extraction of cloud cover features in this paper is based on the grayscale images of the cloud region, it is necessary to extract the cloud region in the ASC images first. Sunlight has a high grayscale value and is very easy to be mistaken for cloud when performing cloud detection in the ASC images. Therefore, it is very important to remove the influence of sunlight. We use the difference method to filter the sunlight in this paper. For a given ASC image, the shooting time and the latitude and longitude of the shooting location are known, so we can use the image with different dates but the same moment as the clear sky background image with the same sun elevation angle as the original image. The grayscale images of two images are first obtained and the difference operation is performed to acquire the image after removing the sun. Then the cloud detection results are obtained using the binarization method. Finally, the cloud region is extracted by applying the image multiplication method.

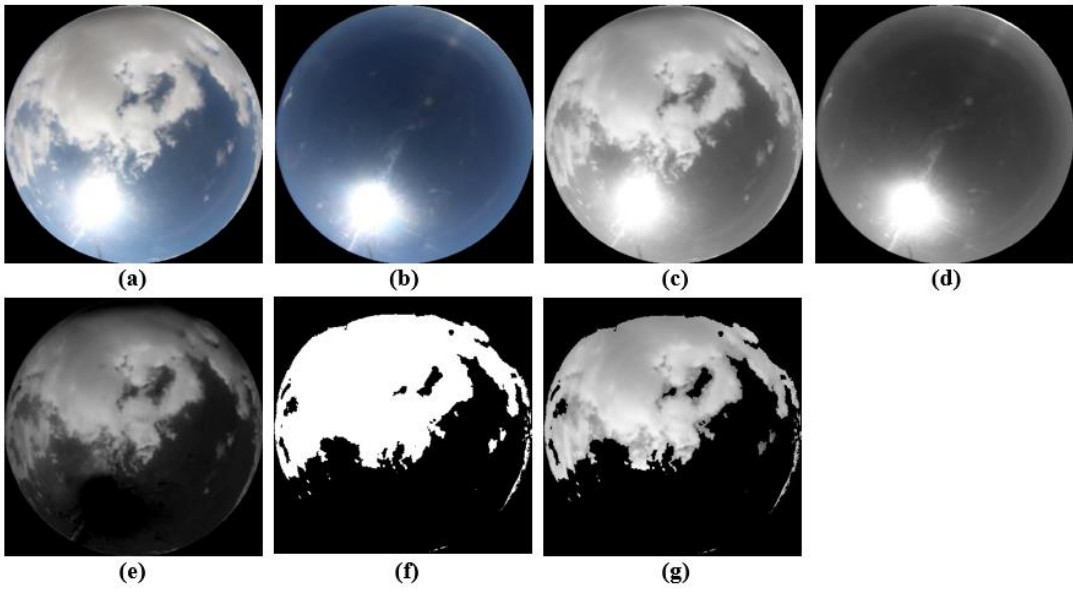



**Figure 6.** Cloud extraction result. (**a**) Original ASC image, (**b**) real clear background image with the same solar elevation
angle as (**a**), (**c**) grayscale image of (**a**), (**d**) grayscale image of (**b**), (**e**) difference of (**c**) and (**d**), (**f**) cloud detection result, (**g**)
cloud extraction result.

Figure 6 shows an example to illustrate the various steps of extracting cloud region. Figure 6a is the acquired original ASC
image and Fig. 6b is the clear sky background image with the same sun elevation angle. Figure 6c and Fig. 6d are the grayscale
images of Fig. 6a and Fig. 6b respectively, which have very similar brightness distributions. Figure 6e is the result of Fig. 6c
minus Fig. 6d, and it can be seen that the influence of the sun has been completely eliminated. The cloud region shown in Fig.
6f can then be obtained by binarization, and the final extracted cloud region (Fig. 6g) can be obtained by multiplying Fig. 6f
with Fig 6c. Generally speaking, the cloud detection accuracy of traditional methods around the sun and near-horizon regions
is relatively low, but the method used in this paper can achieve better results in all regions. For thin cloud regions, this method
can also accurately identify them. In addition, it is possible to exclude bright noises due to light refraction because they are in
the same position and have similar brightness in Fig. 6c and Fig. 6d.

### 3.2.2 Extraction of cloud cover features

In this paper, we proposed three cloud cover features: cloud weight, cloud area ratio and cloud dispersion, which basically
contain the information of cloud coverage area, thickness and distribution location. We represent the computational models of
three features separately, which should theoretically match the TMT classification criteria. In order to improve the accuracy
of ASC image classification, we extracted five features of each image separately, including: cloud area ratio ($CAR$), cloud
dispersion ($CD$), inner circle cloud weight ($w_i$), outer circle cloud weight ($w_o$) and global cloud weight ($w_g$). Among them,
$w_i$ represents the average cloud weight within the inner circle, $w_o$ denotes the average cloud weight between the inner and the
outer circle and $w_g$ indicates the average cloud weight within the outer circle. Besides, the effective range of both $CAR$ and
$CD$ is within the outer circle. The steps of the feature extraction algorithm are shown in Algorithm 1:

---

**Algorithm 1:Extraction of cloud cover features**

**Input:** preprocessed ASC images and corresponding clear sky background images

**Output:** feature vector $v = (CAR, CD, w_i, w_o, w_g)$

1: Calculate the relative distance between each pixel and the image center $D(i,j)$, where i and j represent the
pixel coordinates

2: Compute the true grayscale value of the cloud pixel $G_{cloud}(i,j)$ according to Eq. (4)

3: Calculate the value of influence degree function $f[D(i,j)]$ using Eq. (12)

4: Obtain the $w$, $CAR$ and $CD$ of cloud pixels according to Eq. (8), (10) and (13)

5: Count the total number of cloud pixels within the inner circle and between the inner and outer circle, calculate
the $w_i$, $w_o$ and $w_g$

---



Table 2 shows examples of the value of cloud cover features for each type of ASC images, and the value are normalized for the convenience of comparison. As can be seen from the table, the size of the cloud weight is determined by the thickness of the cloud in the specified area, so there is no obvious difference between the cloud weight of each type image. However, it has a great influence on $CAR$ and $CD$, and the size of cloud weight in the inner and outer circle affects the classification of ASC images based on TMT classification criteria, so the $w_i$, $w_o$ and $w_g$ are used as features for classification in this paper. The $CAR$

and $CD$ are derived on the basis of cloud weight and the two have an overall positive correlation. Besides, there are obvious differences between the values of different categories. Figure 7 shows the distribution of $CAR$ and $CD$ for the four ASC images. As Fig. 7 illustrates, the distribution of feature values of each type ASC image is concentrated in a certain range. The $CAR$ and $CD$ of "Clear" and "Covered" have a more obvious division in values, while the overlap phenomenon exists in "Inner" and "Outer". The distribution ranges of $CAR$ and $CD$ are relatively similar for each class ASC image, but the overlapping part of

$CD$ for "Inner" and "Outer" is reduced compared with $CAR$, which indicates that considering the information of cloud location distribution can distinguish the image classes more effectively. Therefore, the $CAR$ and $CD$ can be used as features to classify the ASC images.

**Table2.** Examples of cloud cover features for each type of ASC images.

| | class | CAR | CD | $w_i$ | $w_o$ | $w_g$ |
|---|---|---|---|---|---|---|
| | Clear | 0 | 0 | 0 | 0 | 0 |
| | Outer | 0.0876 | 0.1169 | 0 | 0.8734 | 0.8734 |
| | Inner | 0.3711 | 0.3230 | 0.7254 | 0.8428 | 0.7963 |
| | Covered | 0.7470 | 0.7834 | 0.6261 | 0.6724 | 0.6459 |





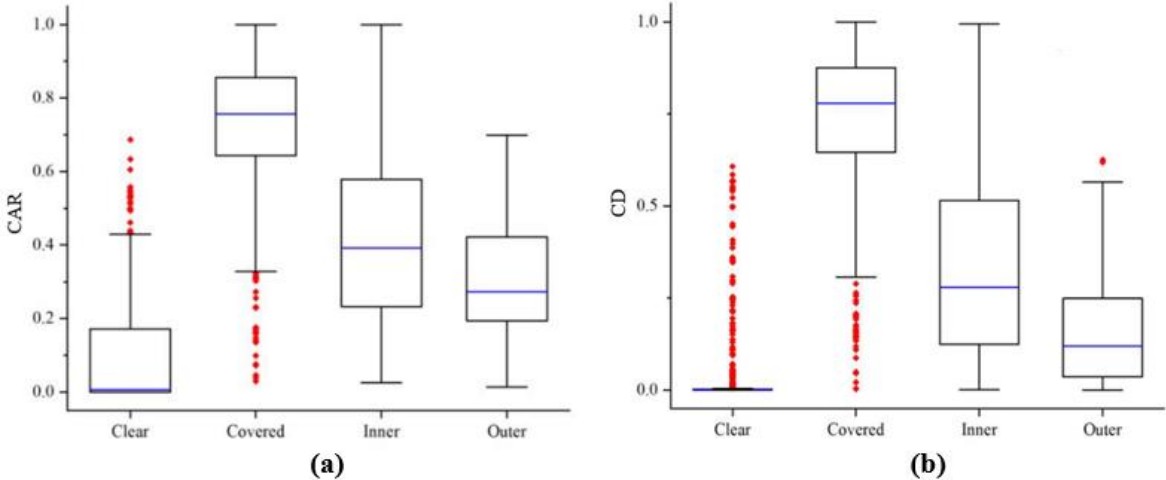


**Figure 7.** Distribution of feature value for each type of ASC image. (**a**) Distribution of CAR, (**b**) distribution of CD.

### 3.2.3 Selection of classifier and training sample

The above features are integrated into a feature set to feed into the classifier for training. To test the effectiveness of our proposed cloud cover features for classification, we selected four classifiers: decision tree (DT), support vector machine (SVM),

K nearest neighbor (KNN) and random forest (RF). There are a total of 7328 different types of ASC images in the data set, and we selected 5863 images for training the classifier and 1465 images for testing the effectiveness.

## 4 Results and discussion

In this section we evaluate the effectiveness of cloud cover features for the classification of ASC images. Then we analyze and discuss the classification results.

**4.1 Experiment results**

The data set is trained and tested using the method in the previous section. To evaluate the classification performance more comprehensively, we use Accuracy, Precision and Recall as evaluation metrics. The accuracy can be calculated based on positive and negative classes as:

$$Accuracy = \frac{TP + TN}{TP + TN + FP + FN} \tag{14}$$

where TP (True Positive) is the number of correctly classified instances for a specific class, TN (True Negative) is the number of correctly classified instances for the remaining types, FP (False Positive) is the number of misclassified instances for the remaining types and FN (False Negative) is the number of misclassified instances for a specific class. And the precision and recall can be expressed as:



$$Precision = \frac{TP}{TP + FP} \qquad (15)$$


$$Recall = \frac{TP}{TP + FN} \qquad (16)$$

In addition, we use F1_score in the evaluation, which can be expressed as:

$$F1\_score = \frac{2 \times Precision \times Recall}{Precision + Recall} \qquad (17)$$

The final test results of the four classifiers are shown in Table 3. It can be seen that the average accuracy of each classifier is more than 95%, indicating that the cloud weight, cloud area ratio and cloud dispersion proposed are effective cloud cover

features that can classify ASC images based on the TMT classification criteria, which greatly promotes the automatic processing of the images. The precision and recall of "Clear" and "Covered" are greater than 95%, while "Outer" and "Inner" are both less than 95%, indicating that "Outer" and "Inner" are easy to be misclassified. Among the four classifiers, the best performer is RF, which has an average accuracy of 97.28% and F1_score of 96.97%.

**Table3.** Comparison of results of four classifiers.

| Classifier | Recall | | | | Precision | | | | Accuracy | F1_score |
|---|---|---|---|---|---|---|---|---|---|---|
| | Clear | Outer | Inner | Covered | Clear | Outer | Inner | Covered | | |
| SVM | 0.9638 | 0.9336 | 0.9528 | **0.9917** | 0.9755 | 0.9328 | 0.9529 | **0.9904** | 0.9650 | 0.9679 |
| KNN | **0.9875** | 0.9312 | 0.9454 | 0.9807 | 0.9658 | 0.9433 | 0.9358 | 0.9902 | 0.9598 | 0.9603 |
| DT | 0.9726 | 0.9228 | 0.9426 | 0.9795 | 0.9687 | 0.9296 | 0.9378 | 0.9857 | 0.9538 | 0.9525 |
| RF | 0.9869 | **0.9406** | **0.9543** | 0.9892 | **0.9926** | **0.9452** | **0.9534** | 0.9882 | **0.9728** | **0.9697** |

**4.2 Performance discussion**

To further analyze the misclassification, we obtained the confusion matrix of random forest classifier as shown in Table 4. It can be seen from the table that the classification accuracy of "Covered" and "Clear" is higher, while "Outer" and "Inner" is lower. The non-zero values of the non-diagonal elements in the table represents the probability of misclassification between classes. By looking at the misclassified images, it can be learned that some "Outer" images are misclassified as "Clear" or

"Inner". Because some "Outer" images have clouds in the inner circle, but the thickness is extremely small, they will be misclassified as "Inner". Or there are only scattered thin clouds in the outer circle, so they are misclassified as "Clear". "Inner" images are also misclassified as "Outer" or "Covered". Some "Inner" images have clouds in the inner circle, but an incorrect classification is caused by the thickness, or some "Inner" images have a thin cloud thickness althought the distribution of clouds is wide, so it is easy to make a misjudgement. Figure 8 displays some misclassified ASC images. The reason for

misclassification is that the thickness of some cloud regions is incorrectly identified. Although the cloud cover features we proposed has taken into account the thickness of the cloud, there is still room for further improvement.





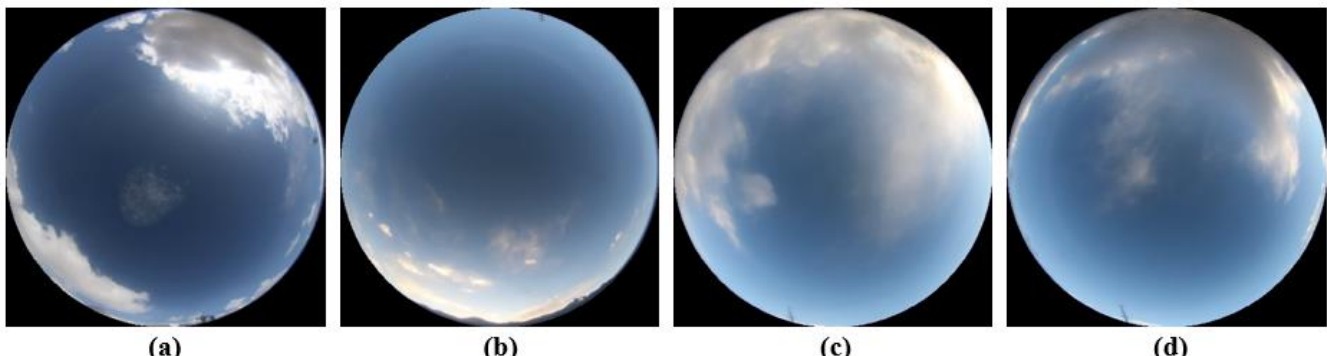

**Figure 8.** Some misclassified ASCimages. (**a**) "Outer" to "Inner", (**b**) "Outer" to "Clear", (**c**) "Inner" to "Covered", (**d**) "Inner" to "Outer".

**Table4.** Confusion matrix of classification result using random forest.

| Ground truth | Classified | | | |
|---|---|---|---|---|
| | Clear | Outer | Inner | Covered |
| Clear | 0.9869 | 0.010 | 0.0031 | 0 |
| Outer | 0.0200 | 0.9406 | 0.0394 | 0 |
| Inner | 0 | 0.0284 | 0.9543 | 0.0173 |
| Covered | 0 | 0 | 0.0108 | 0.9892 |

**5 Conclusions**

This paper proposes three cloud cover features according to the TMT classification criteria, namely cloud weight, cloud area ratio and cloud dispersion, and completes the classification of ASC images based on these features. In this method, the cloud weight indicates the thickness of the clouds, the cloud area ratio represents the distribution range of the cloud and the cloud dispersion reflects the cloud influence degree on astronomical observation results. We quantify these features and then use a classifier to identify classes of ASC images. A large data set is composed of ASC images taken by the all-sky camera located
in Xinjiang, China (38.19° N, 74.53° E), and evaluated to verify the effectiveness of the method. The experiment results show that the highest classification accuracy is 97.28% and F1_score is 96.97% by using the cloud cover feature. Based on this method, astronomical observatory site selection experts can greatly reduce the time to classify the ASC images which will also greatly improve the efficiency of image processing. With comprehensive statistical data, they can choose the best site.



However, this method still has some shortcomings that need to be improved. The classification accuracy of "Inner" and "Outer"
images needs to be increased, and better classification algorithms can be studied for these two types. The sunlight in ASC
images affects the brightness of clouds, and how to better eliminate the influence of the sunlight is also a problem that needs
further research. In addition, this paper does not study the classification of images in bad weather such as rain and fog, and we
will study new classification algorithms with such images in the future.

*Data availability.* The data are provided by the National Astronomical Observatories, Chinese Academy of Sciences, (NAO-
CAS) and we need to apply to the NAO-CAS before we publish the dataset. The data will be made available from author
Xiaotong Li (865517454@qq.com) and corresponding author Bo Qiu (qiubo@hebut.edu.cn) upon request.

*Author contributions.* XL and BQ conceived and designed the study; XL performed the experiments and drafted the manuscript;
BQ and CW proposed constructive suggestions on the revision of the manuscript.

*Competing interests.* The authors declare that they have no conflict of interest.

*Acknowledgements.* This research has been supported by the Joint Research Fund in Astronomy through cooperative agreement
between the National Science Foundation of China (NSFC) and Chinese Academy of Sciences (CAS) under Grant (U1931134).

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
