# Peer review of "An all-sky camera images classification method using cloud cover features"

_Atmospheric Measurement Techniques, 2021_

## Referee Comment (RC1)

As far as I am concerned, in this paper, authors improve the TMT classification Telescope by considering the cloud thickness and density of distribution and then use 4 CNN method to classify 7382 ASC images taken by an all-sky camera. However, I find the manuscript is not ready for publication in its current form and should be returned to authors for major revisions. Deficiencies are listed below:

**Major Point:**

1. The advantage of authors' improvement should also be highlighted and investigated.

(a) What is different after considering the cloud thickness into the classification. To what degree can this adjustment improve? A robust comparison between classification of considering the cloud thickness and ones of not considering the cloud thickness is necessary. In addition, is there any good classification results of ASC image which can be used for benchmark value to investigate the author's improvements?

(b) I suggest authors shorten the comparison between 4 CNN methods and give the above comparison instead.

2. The verification process of Eq. (3) should be added in the revised manuscript.

(a) There are so many expressions between $\alpha$ and $G_{cloud}$ which accord with the principle in line 123 to 127. I believe if author choose another expression (e.g. $\alpha = G_{sky}/G$), the classification results are completely different.

(b) From Fig. 4, I see some thin clouds are also removed, to what degree is it affect final classification results.

3. I also doubt the validation of Eq. (6). Following the radiative transfer theory, the transmittance can be written as:

$$t = e^{-\tau},$$

where T is the transmittance and $\tau$ is optical depth which can use cloud thickness represent. Therefore, the expression between the reflectivity and the cloud thickness is

$$r = \lambda(1 - e^{-w}).$$

**Minor Point:**

1. The number of Eq. In section 3.1.3 may be mistaken.

2. The figure caption should be more detailed. For example,

(a) The information of Table. 1 can be merged into the caption of Fig. 2.

(b) In the caption of Fig. 4, what is "superimposed model" ? I cannot find it in the text.

(c) The means of $d^*_i$ and $f(d^*_i)$ should be added.

A good figure caption can help reader understand the author's contributions without reading the full paper.

---

## Author Comment (AC1)

**Replies to comments of Reviewer 1**

Authors would like to express sincere thanks to an anonymous reviewer for his/her valuable comments and suggestions. We carefully revised the manuscript following the given suggestions and comments. Our replies to the comments and suggestions are given below.

1. The advantage of authors' improvement should also be highlighted and investigated.

(a) What is different after considering the cloud thickness into the classification. To what degree can this adjustment improve? A robust comparison between classification of considering the cloud thickness and ones of not considering the cloud thickness is necessary. In addition, is there any good classification results of ASC image which can be used for benchmark value to investigate the author's improvements? (b) I suggest authors shorten the comparison between 4 CNN methods and give the above comparison instead.

As suggested by the reviewer, we tested the classification effect of other methods for cloud image classification on this dataset and compared with our method. Specific results are in the revised manuscript (Page 14, Line 306).

2. The verification process of Eq. (3) should be added in the revised manuscript.

(a) There are so many expressions between  $\alpha$  and Gcloud which accord with the principle in line 123 to 127. I believe if author choose another expression (e.g.  $\alpha = G_{sky}/G$ ), the classification results are completely different. (b) From Fig. 4, I see some thin clouds are also removed, to what degree is it affect final classification results.

We derived an expression for the relationship between  $\alpha$  and  $G_{Cloud}$  based on the relationship between the two and ensuring the numerical soundness, and verified the feasibility of the method using Figure 4. During this process, we also considered many other expressions, including your proposed  $\alpha = G_{sky}/G$ . If  $\alpha = G_{sky}/G$ , then  $G_{sky}/\alpha = \alpha G_{sky} + (1-\alpha)G_{cloud}$  according to Eq. (2). We can get  $G_{cloud} = G_{sky} + (1/\alpha)G_{sky}$  according to mathematical knowledge, but there can not be a relationship between  $G_{cloud}$  and  $G_{sky}$  the above relationship. Therefore, the speculation is not valid.

Some very thin clouds are neglected in Fig. 4, and a comparison with the original image shows that this part of the cloud is extremely thin and therefore disappears after processing according to Eq. (4). This part does not affect the telescope observation for the TMT classification criteria, so it has almost no effect.

3. I also doubt the validation of Eq. (6). Following the radiative transfer theory, the transmittance and be written as:  $t = e^{-t}$  where T is the transmittance and  $\tau$  is optical depth which can use cloud thickness represent. Therefore, the expression between the reflectivity and the cloud thickness is  $r = \lambda(1 - e^{-w})$ .

Thank you very much for the challenge. After reviewing the information, we learned that your suggestion is more effective, so we have adopted your suggestion. And the experiment has been rerun in the revised manuscript.

**Minor Point:**

The number of Eq. In section 3.1.3 may be mistaken.

Thank you very much for pointing the mistake. The mistake is corrected in the revised manuscript.

- 2. The figure caption should be more detailed. For example,
- (a) The information of Table. 1 can be merged into the caption of Fig. 2.
- (b) In the caption of Fig. 4, what is "superimposed model" ? I cannot find it in the text.
- (c) The means of d\*i and f(d\*i) should be added.

Thank you very much for your suggestions. We have made changes as you suggested in the revised manuscript.

---

## Author Comment (AC2)

**Replies to comments of Reviewer 2**

Authors would like to express sincere thanks to an anonymous reviewer for his/her valuable comments and suggestions. We carefully revised the manuscript following the given suggestions and comments. Our replies to the comments and suggestions are given below.

Specific comments

In my opinion, the central research topic addressed in this paper was recognition of single pixels cloudy or cloud-free. Performance of all the other findings or suggestions are subject to that. As to recognition, the central idea was to compare (subtract/correlate) each routinely measured image with a clear sky an image obtained at the same date (ie. with the same position of the sun). As we are essentially discussing automated classification here, it remained unclear to me how these cloudless images are routinely obtained? And how does the algorithm perform if there is cloud near but not (fully) overlapping the sun? (I wondered if that area should be outright regarded useless and left out in calculations,instead.) But you make a strong claim! (Line: 232: "Generally speaking, the cloud detection accuracy of traditional methods around the sun and near-horizon regions is relatively low, but the method used in this paper can achieve better results in all regions.")

We first obtain the cloud features of ASC images and then perform the classification in this paper. During this process, features can be extracted directly when there is no sun in the image. When the sun is present, we use the difference method to remove the sun from the image. The difference method is a common method for removing the sun in the field of cloud image processing and has been used in several articles. Each cloud image records the shooting time during the shooting process, so the corresponding cloud-free images can be artificially selected according to the shooting time. In addition, during the selection of the dataset, we discard those images where the sun partially or completely overlaps with the cloud. Therefore, when the sun is present in the image, the sun and clouds do not overlap each other, and the sun can be removed using the difference method.

Technical Corrections

Line 16: The starting sentence (!) contains the word "polymer", a term related to chains of molecules? Speaking about clouds, I wonder if you actually thought of *aeresols* instead?
Line 65: Finally, *it is our conclusion* in Section 5.

Thank you very much for pointing the mistake. The mistake is corrected in the revised manuscript.

Figure 4. The left-hand image (a) is claimed to be original, but looks suspiciously thresholded. The other one (b) looks more like an original. Maybe I just did not understand.

The left-hand image (a) of Figure 4 is the grayscale image after we extract the cloud region using the threshold segmentation method. The purpose of using the threshold segmentation is to remove the pure sky background region from the image, leaving only the region where the clouds are mixed with the sky background. As can

be seen from Fig. 4, the grayscale value of the thin clouds in the processed grayscale image is reduced because the effect of the sky background on the cloud grayscale value is eliminated by the method in this paper.